# Evaluating Focal Areas of Signal Intensity (FASI) in Children with Neurofibromatosis Type-1 (NF1) Treated with Selumetinib on Pediatric Brain Tumor Consortium (PBTC)-029B

**DOI:** 10.3390/cancers15072109

**Published:** 2023-03-31

**Authors:** Natasha Pillay-Smiley, James Leach, Adam Lane, Trent Hummel, Jason Fangusaro, Peter de Blank

**Affiliations:** 1Cancer and Blood Diseases Institute, The Cure Starts Now Foundation Brain Tumor Center, Cincinnati Children’s Hospital Medical Center, University of Cincinnati College of Medicine, Cincinnati, OH 45229, USA; 2College of Medicine, University of Cincinnati, Cincinnati, OH 45229, USA; 3Department of Radiology, Cincinnati Children’s Hospital Medical Center, Cincinnati, OH 45229, USA; 4Children’s Healthcare of Atlanta and Aflac Cancer Center, Atlanta, GA 30322, USA; 5Children’s Healthcare of Atlanta and Emory, University School of Medicine, Atlanta GA 30322, USA

**Keywords:** NF1, FASI, low grade glioma, MEK inhibitor, selumetinib

## Abstract

**Simple Summary:**

Focal areas of signal intensity (FASI) are common neuro-imaging abnormalities in children with neurofibromatosis type 1 (NF1). They may confound tumor evaluations and have been associated with neurocognitive differences in some studies. Selumetinib is a MEK inhibitor recently studied in NF1-associated low-grade glioma by the Pediatric Brain Tumor Consortium (PBTC). The current study looked at the impact of selumetinib on FASI. Unlike its effect on NF1-associated LGG, selumetinib did not change the overall size of FASI in children with NF1.

**Abstract:**

Background: Understanding the effect of selumetinib on FASI may help elucidate the biology, proliferative potential, and role in neurocognitive changes for these NF1-associated lesions. Methods: Patients with NF1-associated LGG and FASI treated with selumetinib on PBTC-029B were age-matched to untreated patients with NF1-associated FASI at Cincinnati Children’s Hospital Medical Center. Paired bidirectional measurements were compared over time using nonparametric tests. Results: Sixteen age-matched pairs were assessed (age range: 2.8–16.9 years, 60% male). Initial FASI burden was not different between groups (median range 138.7 cm^2^ [88.4–182.0] for the treated subjects vs. 121.6 cm^2^ [79.6—181.9] for the untreated subjects; *p* = 0.98). Over a mean follow-up of 18.9 (±5.9) months, the LGG size consistently decreased with treatment while no consistent change among the treated or untreated FASI size was seen. At the paired time points, the median treated LGG decreased significantly more than the treated FASI (−41.3% (LGG) versus −10.7% (FASI), *p* = 0.006). However, there was no difference in the median size change in the treated versus untreated FASI (−10.7% (treated FASI) versus −17.9% (untreated FASI), *p* = 0.08). Among the treated subjects, there was no correlation between the change in LGG and FASI (r = −0.04, *p* = 0.88). Conclusions: Treatment with selumetinib did not affect the overall FASI size in children with NF1 treated for progressive low-grade glioma.

## 1. Introduction

Neurofibromatosis type 1 (NF1) is a common cancer predisposition syndrome affecting approximately 1:2500 to 3000 people worldwide [1]. Acquired either by spontaneous mutation or inherited by an autosomal dominant pattern, it is caused by a germline loss of function mutation in the NF1 tumor suppressor gene that encodes neurofibromin, a negative regulator of the RAS activity within the MAP kinase pathway [2,3]. NF1 has a heterogenous clinical phenotype, predisposing to a range of NF1 manifestations including the central nervous system (CNS) and peripheral nerve tumors as well as cognitive and behavioral deficits. Low-grade gliomas are the most common CNS tumors seen in children with NF1, with optic pathway low-grade gliomas seen in 15–20% of all children with NF1 and comprising 66–75% of CNS tumors in this population [4,5].

Focal areas of signal intensity (FASI), previously called unidentified bright objects (UBO), are also common findings in the neuro-imaging of patients with NF1 [4]. These areas of T2 hyperintensity are frequently found in the cerebellum, brain stem, basal ganglia and thalami of young children with NF1 [5] and resolve by late adolescence [5,6,7]. While considered benign, they may cause confusion if they are misidentified as tumors [8,9] or if they obscure tumor borders with which they overlap. Low-grade gliomas have developed within FASI [10], leading to the question of whether these lesions have the potential for transformation [4]. FASI in the thalamus have also been associated with cognitive changes in some studies, although most studies show no effect in other brain regions [7,11].

Oral selective MEK inhibitors have offered new treatment options in NF1-associated low-grade glioma (LGG). In PBTC-029, Fangusaro et al. demonstrated that selumetinib decreased tumor dimensions and controlled tumor growth during treatment in the vast majority of children with NF1-associated LGG [2]. Importantly, other studies have suggested that children with NF1 treated with either selumetinib or trametinib (another MEK 1/2 inhibitor) may show improvement in tests of executive function, processing speed, and verbal comprehension [12]. Because of concerns raised over the potential impact of FASI, we sought to determine whether treatment with selumetinib affected the FASI size in children with NF1. We retrospectively compared the change in FASI size to the change in tumor size among children treated with selumetinib on stratum 3 (children with NF1-associated LGG) of PBTC-029. Since FASI size may enlarge or reduce over time and these lesions disappear by adulthood [5,13], we also compared the FASI change to an age-matched cohort of untreated children with NF1 at the Cincinnati Children’s Hospital Medical Center.

## 2. Materials and Methods

PBTC-029B is an IRB approved phase II study conducted by the Pediatric Brain Tumor Consortium (PBTC). Children from 3 to 21 years of age, with recurrent, refractory, or progressive low-grade glioma were enrolled and placed into separate strata based on NF1 status, tumor histology, tumor location, and BRAF aberration status. Stratum 3 enrolled children with NF1 and low-grade glioma (WHO Grade I or II) of any location. Patients were treated with selumetinib (AZD6244), a selective oral small molecule inhibitor of MEK 1/2 for up to 26 courses (approximately 2 years), depending on the tumor response [2,14,15]. Patients with at least one FASI on their baseline MRI were eligible for the current study. Imaging to assess tumor change was performed every two months until course 10, and then every three months for the duration of therapy.

Children with NF1 seen at the Cincinnati Children’s Hospital Medical Center (CCHMC) receive imaging at approximately 2 years of age to screen for NF1-associated LGG [16]. Untreated subjects in this study included children with NF1-associated LGG that did not require tumor-directed treatment but were followed by surveillance imaging alone. Subjects were eligible to act as controls if they had at least one FASI and underwent surveillance imaging for over 1 year without receiving tumor-directed therapy during or preceding that time. Untreated control subjects were matched to treated study subjects if their ages were within a year of each other at the time of imaging. If multiple age-matched control subjects were available, the patients with the most recent imaging were selected.

A single paired time point was used to assess change over time. Longitudinal imaging was considered paired for treated and untreated subjects if the time from baseline was within 4 months of each other. If multiple scans were eligible to be paired, the longest interval was chosen.

FASI were defined, in accordance with prior studies [4,5,6,17,18,19], as regions of T2 hyperintensity without mass effect, enhancement, or T1 hypo-intensity relative to gray matter. One FASI from each of the following five areas was evaluated: cerebellum, lentiform nucleus, thalamus, brainstem, and cerebral hemispheres. Lesions were evaluated on T2 weighted imaging and the product of their longest diameter and longest perpendicular diameter were followed longitudinally. Because lesions in multiple areas may be measured, the sum of the cross products was used to estimate the total burden of FASI. All FASI were assessed and measured by a pediatric neuro-radiologist (JL) who also determined whether they qualified as FASI by the definition stated above. LGG were similarly measured as the product of perpendicular dimensions as reported in the study.

### Statistical Analysis

The LGG size and cumulative FASI burden were described in all subjects. Change in these measures was plotted at all available time points to describe the tumor and FASI burden over time. The change in lesions between the baseline and the latest paired time point was described graphically and compared using a Wilcoxon paired sign-rank test. Responses at these time points were described as complete response (CR, no evidence of lesion), partial response (PR, 50% or greater reduction in lesion cross product), stable disease (SD, change between less than 50% decrease and 25% increase), or progressive disease (PD, greater than 25% increase in lesion cross product), which matched the response criteria used for PBTC-029 [2]. The Fisher exact test was used to compare the proportions for each category (treated LGG, treated FASI, untreated FASI). The Spearman’s rank correlation was used to examine the relationship between the changes in FASI and changes in tumor size.

Since FASI limited to the thalamus and basal ganglia have been implicated in neurocognitive differences among children with NF1, a sub-analysis separately examined FASI in these areas.

## 3. Results

Of the 25 subjects enrolled on stratum 3 of PBTC-029, nine had no evidence of FASI on their baseline pre-treatment scan. For the remaining 16 selumetinib treated subjects, age-matched untreated controls were identified from children with NF1 followed at CCHMC. Age, gender, and initial FASI burden for each pair is shown in Table 1. Baseline age ranged from 2.9 to 16.6 years old, a mean age of 10.7 years old for the treated subjects, and 10.2 years old for the untreated controls. Patients were not matched for gender; however, 9/16 (56%) of both groups were male. A total of 33 FASI were measured among the treated subjects (mean 2.1 FASI/subject), and 32 FASI were measured among the untreated controls (mean 2 FASI/subject). No FASI later developed changes in enhancement, mass effect, or signal characteristics over the course of the study. Among both the treated and untreated subjects, the cerebellum was the most common site for FASI: 15/16 subjects in both the treated and untreated groups had measurable FASI in the cerebellum. The lentiform nuclei (seven FASI among treated subjects, 12 FASI among untreated controls) and the brainstem (seven FASI among treated subjects, two FASI among the untreated controls) were also common locations. The initial burden of FASI, defined as the sum of measured FASI cross-products, was not different between the control and subject group (*p* = 0.98; median [IQR]: 138.7 cm^2^ [88.4–182.0] for the treated subjects vs. 121.6 cm^2^ [79.6–181.9] for the untreated controls).

Changes in the lesions over time was demonstrated graphically by spider plots in Figure 1. Mean follow-up time was 18.9 (±5.9) months. Percent change in the cross product from the baseline is shown for LGG as well as the treated and untreated FASI. Since the FASI size may change over time independent of treatment [5], it is important to compare the treated FASI to the untreated control FASI. FASI in both the treated and untreated groups demonstrated diverse responses over time, both increasing and decreasing in size without a consistent trajectory (Figure 1a,b). No consistent differences were seen between the treated and untreated FASI. In contrast, the size of the LGG lesions consistently decreased after starting selumetinib, and often retained a similar decrease in size throughout the course of therapy (Figure 1c).

Waterfall plots assessed the tumor and FASI responses to treatment at the paired time points (Figure 2). There was no match available for one pair, so 15 treated/untreated pairs were assessed for paired longitudinal analysis. The average follow-up time for the paired analysis was 18.5 (±6.5) months. At the paired time points, the median change in FASI exposed to treatment was −10.7% [range: −26.3–12.9] compared to −17.9% [range: −28.0–9.0] among the untreated subjects (*p* = 0.08 Wilcoxon signed rank test). At the paired time point, 10/15 treated subjects had a decrease in FASI size compared to 14/15 untreated control subjects. This difference was not statistically significant. Among the treated subjects, there was a statistically significant difference in response to FASI (median −10.7% [max-min: −26.3–12.9]) compared to LGG (median −41.3% [max-min: −95.6–3.6]) at the paired time point (*p* = 0.006). There was no correlation between the change in LGG and change in FASI at the paired time points (r = −0.04, *p* = 0.88) (Figure 3). Figure 4 shows the change in FASI and tumor in a single patient over time.

Responses for FASI and LGG were assessed using the response criteria defined above at the paired time points. The results are found in Table 2. The majority of FASI were stable over this time period (12/15 treated FASI vs. 14/15 untreated FASI, Fisher exact test *p* = 0.73). In contrast, 7/15 LGG demonstrated a partial response to selumetinib (*p* = 0.008 compared to FASI).

Given the differences in cognition cited in the literature in children with FASI in the thalamic region [20,21,22] we performed a sub-analysis of treated and untreated patients with FASI in the thalamic and lentiform regions of the brain. There were 7 treated patients and 14 untreated patients with FASI in this region. The median change in the FASI among treated subjects was −0.48% (range: −27.36–253.59). The median change in the FASI among untreated subjects was −16.01% (range: −57.24–23.67). A spider plot (Figure 5) shows that most untreated FASI decreased in bi-directional cross product, while a few treated FASI increased substantially in size. At paired time points, there were six paired subjects. In each of the six pairs, the outcome for thalamic or lentiform FASI after treatment was worse (more growth or less shrinkage) than without treatment. In a Wilcoxon signed rank test, the change in cross product for untreated FASI was less than the change in cross product for their treated counterpart (*p* = 0.03). Thalamic FASI in patients not treated with selumetinib changed more (decreased, as expected with FASI over time) than those that were treated. It is uncertain whether selumetinib has a causative effect on the stability of the FASI, given the small sample size of this study.

## 4. Discussion

FASI are common neuroimaging findings in children with NF1. Despite their frequency, there have been few pathologic studies describing these lesions due to their benign nature. One of the few pathologic studies of FASI described glial proliferation with spongiotic or vacuolar change without inflammation, demyelination, or axonal abnormality [4]. This contrasts with glial lesions with cellular atypia found in NF1-associated LGG [15,23]. Fluid-filled vacuoles have been proposed as one reason for the T2 hyperintensity found in FASI, but it is unclear why this hyperintensity may decrease in size and intensity with time. FASI have been the subject of multiple radiographic studies in NF1, but the effect of MEKi on FASI were previously unknown.

Radiographically, FASI have been defined as areas of T2 hyperintensity without contrast enhancement, mass effect, or diffusion restriction [5,6,19,24]. In children with NF1, FASI may sometimes resemble LGG, since LGG can occur outside the optic pathway, is frequently T2 hyperintense, and often do not restrict diffusion [25,26]. They are found more frequently in children with the NF1-associated optic pathway glioma [27]. Moreover, the natural history of FASI can mimic LGG during childhood: the incidence of FASI and LGG peak in early childhood, and both may spontaneously resolve with age although this is much less common in LGG [4,25,28]. Previous studies have described LGG developing in areas of FASI [4,10,29] and have suggested that FASI may have a proliferative potential in children with NF1. For instance, Griffiths et al. reported on the FASI characteristics of 46 children with NF1. While the majority of FASI resolved with age, five out of the 46 increased in size and developed enhancement [4]. These areas were not biopsied but were postulated to have either transformed into LGG or were histologically LGG that eventually changed the imaging characteristics. The radiographic similarities in the appearance and timing of either presentation can complicate the identification and measurement of LGGs in children.

In our study, FASI and LGG responded differently to selumetinib exposure. While almost all NF1-associated LGG reduced in size after starting treatment, there was no consistent difference between the treated and untreated FASI. In the longitudinal analysis over an average 1.5 years of treatment, 13/15 LGG demonstrated some reduction in tumor size including seven partial responses. FASI response to treatment was more diverse: 2/15 demonstrated some growth, 1/15 demonstrated some reduction in size, and 12/15 demonstrated stable disease. However, a comparison of the treated FASI to treated LGG at all time points (Figure 1) or at a single time point (Figure 2, Table 2) showed that there was no consistent response among FASI to selumetinib exposure. The moderate, variable effect of selumetinib on FASI was indistinguishable from the change in FASI in the control group, suggesting that this variability is [30] due to the natural history of FASI rather than the effect of selumetinib. Given these findings, we postulate that there is no effect on FASI by MEK inhibition.

The majority of NF1-associated LGG exhibited bi-allelic NF1 inactivation, and previous reports have shown that only 11% have additional mutations [31]. While our analysis does not preclude FASI having a proliferative potential, it seems unlikely that bi-allelic NF1 inactivation underlies FASI development since responses to MEKi differ between FASI and NF1-associated LGG. Based on the available literature, the vast majority of NF1-associated LGG will decrease in size with selumetinib, and thus the response to MEKi may help distinguish FASI from tumors. In patients with NF1-associated LGG that do not respond to MEKi, FASI that overlap the LGG tumor borders may be considered. However, the proposed FASI portion should continue to meet the imaging definition of FASI with T2 hyperintensity and the lack of enhancement, mass effect, and T1 hypointensity.

Although considered benign lesions, FASI have also been implicated in NF1-associated cognitive differences in some studies. Neurocognitive and behavioral challenges are present in up to 80% of children with NF1 [22,32]. Numerous studies have examined the role of FASI in neurocognitive deficits. Although most studies showed no correlation [7,11], FASI in the thalamus has been associated with cognitive deficits in numerous publications with some studies demonstrating that cognitive deficits improve with resolution of thalamic FASI [11,20,21,22,30,33,34,35]. NF1 mouse models treated with MEK inhibition have shown an improvement in cognitive function [6]. Two clinical studies have also demonstrated improvements in cognitive function in children with NF1 exposed to selumetinib or trametinib for plexiform neurofibroma [36,37]. Improvements were noted in working memory, executive function, verbal comprehension, and processing speed.

The cognitive studies above suggest that central gray-matter FASI may be implicated in neurocognitive deficits associated with NF1, and the treatment of FASI with MEKi may help ameliorate these impairments. However, our study did not show a difference in bi-directional cross product between the treated and untreated FASI. While not statistically significant, a greater proportion of untreated FASI actually demonstrated a decreased size compared to the treated FASI at the paired time points (14/15 vs. 10/15; *p* = 0.08). This difference was also seen when we analyzed the FASI located specifically in the thalamus and lentiform: untreated FASI had less growth or a greater decrease in size over time compared to FASI exposed to selumetinib. In this small, unplanned analysis, this difference did meet the statistical significance (*p* = 0.03). These comparisons contradict the theory that MEKi improves cognition by reducing FASI, but future prospective studies with a larger sample size and paired neurocognitive assessments would be valuable. 

This study is subject to important limitations. FASI exhibit variable growth patterns that make measuring the effect of therapies challenging. We compared our treated group to an age-matched untreated group to account for this variability. This retrospective study should be repeated on a larger scale with other MEKi therapies to confirm our findings, ideally in a prospective manner. Evaluating the FASI size by volumetrics instead of bi-directional measurement may also provide greater accuracy in looking at the growth pattern of FASI. Our study lacks concurrent neurocognitive assessments, which would help determine the cognitive effects of FASI and MEKi therapy. Although one strength of the study is the consistency of treatment and analysis of subjects exposed to selumetinib in the clinical study, neurocognitive assessments were not part of this clinical trial. Both LGGs and FASI from subjects treated on PBTC029B and the paired controls were reviewed by a single neuroradiologist to minimize the variability in interpretation.

## 5. Conclusions

FASI are commonly occurring neuroimaging anomalies in children with NF1. They can cause concern when they are mistaken for LGG or other entities, overlap LGG, and obscure tumor borders. They have also been implicated in neurocognitive deficits associated with NF1. Our study demonstrates that MEK inhibition with selumetinib does not affect the NF1-associated FASI size, despite a consistent reduction in the size of NF1-associated LGG treated with selumetinib. The differential effect of selumetinib on FASI compared to LGG may offer an important distinguishing feature between these entities. This study sheds light on the proliferative potential of FASI as well as their role in neurocognitive deficits associated with NF1.

## Figures and Tables

**Figure 1 cancers-15-02109-f001:**
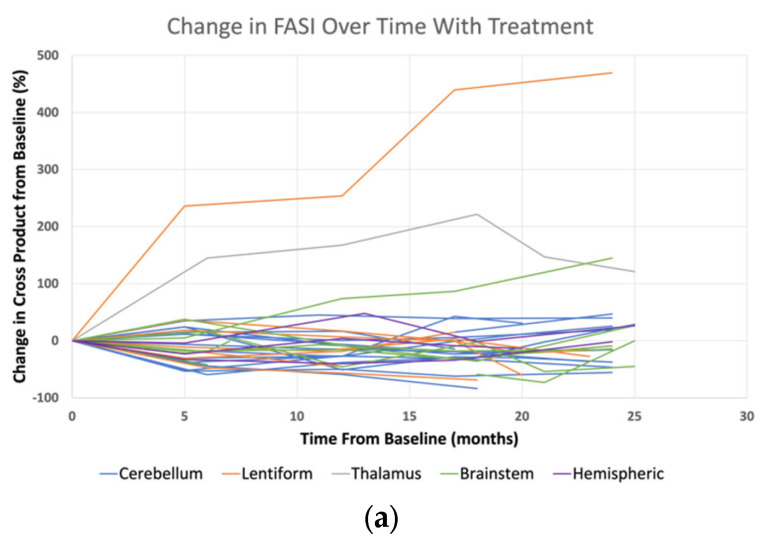
Spider plots demonstrating the percent change in the cross product of lesions from the baseline. Change in FASI from the chosen anatomic locations are shown on treatment with selumetinib (**a**) and without treatment (**b**) with variable responses. Treatment of LGG with selumetinib showed a consistent decrease over time (**c**).

**Figure 2 cancers-15-02109-f002:**
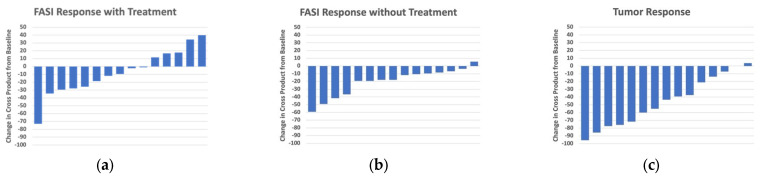
Longitudinal analysis of 15 treated/untreated subjects analyzed at paired time points over an average follow-up time of 18.5 (±6.5) months, with each bar representing individual patients. Decrease in the FASI size with treatment (**a**) compared to without treatment (**b**) did not reach statistical significance. In the treated patients, there was a significant difference between the treated LGG (**c**) and treated FASI (**a**).

**Figure 3 cancers-15-02109-f003:**
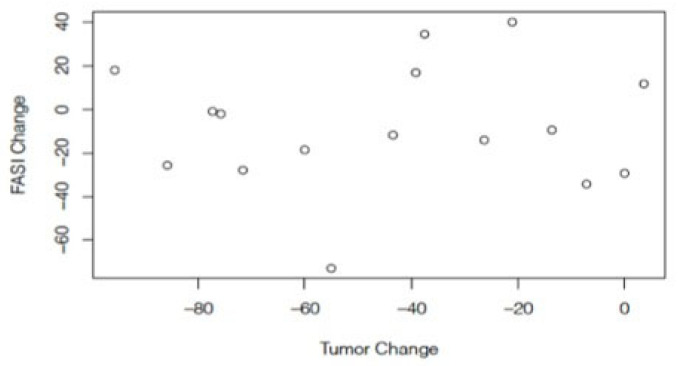
Comparison of the change in FASI to the change in the LGG size at paired time points did not show any correlation.

**Figure 4 cancers-15-02109-f004:**
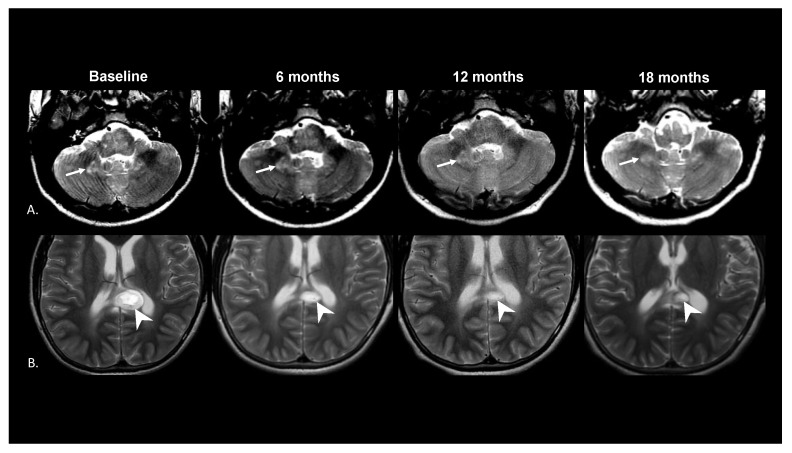
T2 weighted MR imaging of a typical LGG and FASI at four time points during treatment. (**A**) The measured FASI in the right medial cerebellum (arrow) stayed relatively stable during the treatment course. (**B**) The presumed low-grade glioma involving the splenium of the corpus collosum (arrowhead) decreased substantially from the baseline over the treatment course.

**Figure 5 cancers-15-02109-f005:**
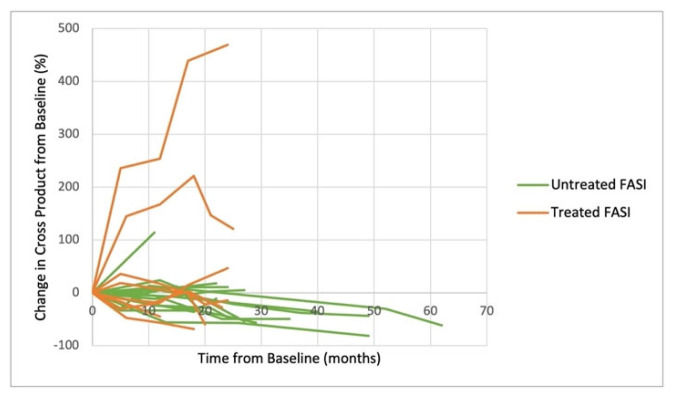
Change from the baseline in the size of the thalamic and lentiform FASI. Spider plot demonstrates the decrease in most untreated FASI with a substantial increase in two treated FASI.

**Table 1 cancers-15-02109-t001:** Demographics of the paired subjects and controls.

Selumetinib Treated Subjects				Untreated Controls			
Pair #	Age at Initial MRI (Years)	Gender	Sum of Cross Products of all FASI (mm^2^)	Pair #	Age at Initial MRI (Years)	Gender	Sum of Cross Products of All FASI (mm^2^)
			Located in Cerebellum Unless Otherwise Specified				Located in Cerebellum Unless Otherwise Specified
1	3.5	Male	72.33	1	2.9	Male	81.13
2	5.7	Male	346.54	2	4.8	Male	67.6
3	6.7	Female	77.2	3	5.7	Male	74.1
4	6.8	Male	134.73	4	6.3	Female	127.56
5	7.3	Male	152.44	5	6.3	Female	411.09
6	7.5	Female	14.4	6	7.3	Female	239.17
**7**	9.8	Male	92.07	**7**	9	Male	92.78
8	10.2	Female	52.52	8	10	Male	134.89
9	11.7	Male	119.1	9	11.1	Male	216.83
10	11.8	Male	159.88	10	11.4	Female	159.3
11 *	12.9	Male	606.48	11*	12.9	Male	105.18
12	13.3	Female	142.65	12	13.8	Male	310.55
13	14.3	Male	240.8	13	14.6	Female	115.6
14	14.3	Female	267.56	14	14.2	Female	21.07
15	15.8	Female	121.06	15	15.7	Female	75
16	16.6	Female	162.33	16	16.5	Male	170.3

* subject removed from paired longitudinal analysis due to the lack of unmatched control within 4 months.

**Table 2 cancers-15-02109-t002:** Response criteria at paired time points for FASI with and without treatment and treated LGG.

	Complete Response	Partial Response	Stable Disease	Progressive Disease
FASI (treated)	0	1	12	2
FASI (untreated)	0	1	14	0
LGG (treated)	0	7	8	0

## Data Availability

The data presented in this study are available on request from the corresponding author. The data are not publicly available due to institutional restrictions.

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
