# Peer review of "Evaluating Focal Areas of Signal Intensity (FASI) in Children with Neurofibromatosis Type-1 (NF1) Treated with Selumetinib on Pediatric Brain Tumor Consortium (PBTC)-029B"

_cancers, 2023, doi:10.3390/cancers15072109_

Round 1

Reviewer 1 Report

This is a nicely presented paper looking at the effects of MEK inhibition on Focal Areas of Signal Intensity (FASI) in patients with Neurofibromatosis type 1 treated on clinical trial PBTC-029 with selumetinib to treat low-grade glioma(s) in these patients vs. matched controls who were not treated with MEKi. Sum of cross product of the measurements of the areas of FASI in each patient were recorded and tracked over time in the setting of the clinical trial and then in the age-matched controls. Results revealed that although the LGGs consistently decreased in size while on MEKi, there were no significant changes in FASI size in these patients. On the contrary, patients not treated with MEKi (control) were more likely to have decrease in size of their FASI compared to MEKi treated patients. Although MEKi has been proposed as a consideration to use in patient with just FASI and not LGG, this study shows, although somewhat limited in patient number, that there is not decrease in FASI when treated with selumetinib.  The methods were described well and the conclusions from the data are appropriate.

I would suggest higher resolution figures for 1a, 1b, and 3 for ease of readability (likely already done, and just how this was submitted) as well as some more development of legends for the figures, especially Figure 2 where it is a bit unclear what is being shown (assuming each bar is a patient as is typical in waterfall plot, but figure description should specify what is being shown).

Overall, very well written and presented study that adds nicely to the literature. 

Author Response

We appreciate the Reviewer's kind feedback of our manuscript. We have replaced the figures in the paper with ones of higher resolution and improved the legends of each figure to be more descriptive. We would also be happy to work with the editors of MPDI to ensure that the figures are of excellent quality and resolution. 

Reviewer 2 Report

Pillay-Smiley et al. present data on an interesting side study of PBTC-029B looking at the effect of selumetinib on NF-1 associated lower grade gliomas. They look at effects of selumetinib in focal areas of signal intensity (FASI) and did not find a universal inhibitory effect on FASI. Age-matched controls from not treated NF-1 patients with FASI were used for comparison. These results suggest that MEK pathway activation due to bi-allelic NF-1 loss are not responsible to formation and growth of FASI. This study is fairly well conducted and its limitations are clearly stated. I only have a couple of minor comments.

1. For figure 4, specific dates should not be presented.

2. In the second to last paragraph of page 8, there needs to be closing bracket after reference 34. ([11,20-22,32,33,34). Also, the paper by Feldmann is not properly referenced as a number (EndNote error).

3. Resolution of figures 1 and 3 should be improved.

Author Response

We appreciate the recommendations of the Reviewer and have corrected the resolution of the figures, the citation error and removed specific dates from Figure 4.